# Simulation Study of High-Precision Characterization of MeV Electron Interactions for Advanced Nano-Imaging of Thick Biological Samples and Microchips

**DOI:** 10.3390/nano14221797

**Published:** 2024-11-08

**Authors:** Xi Yang, Liguo Wang, Victor Smaluk, Timur Shaftan, Tianyi Wang, Nathalie Bouet, Gabriele D’Amen, Weishi Wan, Pietro Musumeci

**Affiliations:** 1National Synchrotron Light Source II, Brookhaven National Laboratory, Upton, NY 11973, USA; vsmalyuk@bnl.gov (V.S.); shaftan@bnl.gov (T.S.); tianyi@bnl.gov (T.W.); bouet@bnl.gov (N.B.); 2Laboratory for BioMolecular Structure, Brookhaven National Laboratory, Upton, NY 11973, USA; lwang1@bnl.gov; 3Physics, Brookhaven National Laboratory, Upton, NY 11973, USA; gdamen@bnl.gov; 4School of Physical Science and Technology, ShanghaiTech University, Shanghai 201210, China; wanwsh@shanghaitech.edu.cn; 5Department of Physics and Astronomy, University of California, Los Angeles (UCLA), Los Angeles, CA 90095, USA; musumeci@physics.ucla.edu

**Keywords:** electron sample interaction, MeV-STEM/TEM, Monte Carlo simulation, angular broadening, biological sample, microchip, detector

## Abstract

The resolution of a mega-electron-volt scanning transmission electron microscope (MeV-STEM) is primarily governed by the properties of the incident electron beam and angular broadening effects that occur within thick biological samples and microchips. A precise understanding and mitigation of these constraints require detailed knowledge of beam emittance, aberrations in the STEM column optics, and energy-dependent elastic and inelastic critical angles of the materials being examined. This simulation study proposes a standardized experimental framework for comprehensively assessing beam intensity, divergence, and size at the sample exit. This framework aims to characterize electron-sample interactions, reconcile discrepancies among analytical models, and validate Monte Carlo (MC) simulations for enhanced predictive accuracy. Our numerical findings demonstrate that precise measurements of these parameters, especially angular broadening, are not only feasible but also essential for optimizing imaging resolution in thick biological samples and microchips. By utilizing an electron source with minimal emittance and tailored beam characteristics, along with amorphous ice and silicon samples as biological proxies and microchip materials, this research seeks to optimize electron beam energy by focusing on parameters to improve the resolution in MeV-STEM/TEM. This optimization is particularly crucial for in situ imaging of thick biological samples and for examining microchip defects with nanometer resolutions. Our ultimate goal is to develop a comprehensive mapping of the minimum electron energy required to achieve a nanoscale resolution, taking into account variations in sample thickness, composition, and imaging mode.

## 1. Introduction

Imaging large and thick biological samples in their native states, such as the nucleus—a complex network of DNA, RNA, and proteins with diameters ranging from 5 to 10 µm [1], presents significant technical challenges that demand advanced microscopy techniques. In this context, the mega-electron-volt scanning transmission electron microscope (MeV-STEM) has emerged as a promising tool capable of accommodating samples with thicknesses exceeding 10 µm [2], though further validation of this capability is necessary. A high-energy MeV-STEM has the potential to overcome the inherent limitations of low-energy electron tomography (cryo-ET), which primarily involves uncertainties and prolonged processing times associated with the cryo-focused ion beam (FIB) slicing of large biological specimens. Traditional cryo-ET techniques can produce only a few 300 nm-thick lamellae per hour, and acquiring a complete high-resolution 3D image of a biological cell may take an entire day [3,4,5,6]. Moreover, charging artifacts from lipid deposits, curtaining effects from density variations, and linear artifacts from the milling process add significant complexity to imaging. Thus, rapid, efficient imaging of thick samples at the nanoscale resolution is essential for advancing scientific discovery [2].

Recent advancements in photocathode guns and superconducting radio frequency (SRF) cavities suggest that constructing a MeV ultrafast electron microscope (UEM) could be feasible within a reasonable budget. Prior simulation studies and ongoing MeV-UEM hardware development indicate that essential components, such as ultra-low emittance photocathode guns, SRF accelerating cavities, and momentum apertures, have already been successfully demonstrated [2]. A MeV-STEM utilizes high-energy electrons (≥3 MeV), which facilitate both elastic and inelastic scattering processes characterized by small critical angles and deep penetration depths [2,7,8,9,10]. This unique attribute is particularly advantageous for STEM imaging modes that focus on amplitude contrast [8], which is essential for resolving intricate biological structures. In contrast, transmission electron microscope (TEM) imaging modes predominantly rely on phase contrast [2], which may not be as effective for such applications.

Optimizing electron beam parameters is crucial for maximizing the performance of MeV-STEM/TEM in imaging large, thick biological specimens [7], examining microchip defects (see Figure A1 in the Section A.1), and achieving nanoscale resolution. Key parameters include electron beam emittance, energy, energy spread, and current density, all of which directly influence imaging resolution and signal fidelity. The effects of angular broadening are significantly more pronounced in thick biological samples compared to sub-micron thin samples. The critical angles for elastic and inelastic scattering depend strongly on electron energy [7], decreasing from 11.84 to 2.14 mrad for elastic scattering and from 0.81 to 0.17 mrad for inelastic scattering as electron energy increases from 300 keV to 3 MeV. Therefore, to maintain a projected beam size on the nanoscale, a minimum electron beam energy is necessary for specific sample thicknesses. Precisely optimizing electron beam energy is crucial for achieving a balance between adequate penetration depth (see Figure A2 in the Section A.2) and high signal fidelity, as well as minimizing the potential damage to biological samples from electron interactions [2].

We conducted simulations over a wide range of beam energies (1 to 10 MeV), using electron beam parameters based on a model proposed by Yang et al. [2] and the PEGASUS accelerator [11,12,13]. The study includes samples with thicknesses ranging from 0.0 to 20.0 µm, chosen to represent biological compositions primarily composed of light elements such as carbon, nitrogen, and oxygen [2,7,8], as well as silicon wafers commonly used in microchip manufacturing.

We simulated transmitted electron intensity profiles and angular distributions [2,7,14], with intensity profiles reconstructed by reverse-propagating the simulated detector signal to the sample exit. This enabled us to analyze beam size, divergence, and the effects of material interactions on electron beam attenuation. Such simulations can address fundamental questions, including potential deviations from the Beer–Lambert law under variable detector collection angles and electron beam energies (see Figure A3 in the Section A.3). The simulations employ a multislice wave optics method (Section A.1) to rigorously model elastic scattering processes, while thermal diffuse scattering is incorporated via the Einstein model, represented as an absorption potential to statistically capture inelastic scattering processes [15,16]. Benchmarking these models against experimental electron transmission data will enable a reliable assessment of their predictive accuracy.

Furthermore, adjusting the drive laser system to manipulate the electron bunch structure will allow us to explore how radiation damage to various biological samples correlates with changes in electron bunch properties, such as structure, energy, and intensity (see Figure A4 in the Section A.4). Beam damage effects present a fundamental limitation on achievable imaging resolution, particularly with MeV electrons. This limitation could negate the advantages of high penetration depth and reduced inelastic scattering backgrounds offered by MeV-TEM imaging, as illustrated in Figure A1c in the Section A.1. This research will also incorporate an energy-resolved characterization of transmitted electrons [17]. These advancements are expected to yield cost-effective designs for MeV-STEM instruments, enhancing imaging resolutions through the optimization of electron beams and focusing parameters for complex biological samples and microchip defects.

## 2. Results

### 2.1. Select Electron Energy

These simulations determine the optimal electron beam energy for a MeV-STEM instrument designed for the nano-imaging of large, thick biological samples. The primary objective is to establish a standardized methodology that reliably provides detailed information across a wide range of sample compositions and thicknesses [18,19,20,21,22,23]. Key requirements include the precise characterization of beam intensity, divergence, and size at the sample exit, with a particular focus on achieving divergence measurements within a few to tens of milliradians, maintaining a precision of a few percent. For thin samples, high resolution relies on tightly focusing the electron beam to sub-nanometer dimensions on the specimen [8,9,10,24,25]. In contrast, when imaging thick biological samples using MeV-STEM, the resolution is predominantly influenced by angular broadening (AB) as electrons traverse the specimen [2,7]. AB accounts for the convergence, semi-angle of the incident beam, and the broadening scattering resulting from the angular distribution induced by all scattering events, including both single and multiple elastic and inelastic scatterings [7]. This phenomenon can be quantitatively assessed by measuring divergence at the sample exit.

Our initial investigations show the significant impact of electron beam energy on AB [7]. To minimize the AB effects as electrons traverse the sample, precise control over the accelerating voltage of the electron beam is crucial, adjustable within the range of 1 to 10 MeV. This flexibility facilitates the determination of the optimal electron beam energy to mitigate AB effects. The electron source utilized in our study builds upon advancements from previous MeV-STEM research, featuring an exceptionally low emittance of 2 picometers [2]. This source, combined with optimized STEM column optics, enables a precise electron beam focusing onto the sample, achieving a transverse size of 1 nanometer and a convergence semi-angle of 1 milliradian [2,7]. In an ideal scenario where the electron beam is precisely centered within the sample thickness, the maximum projected column size of the electron beam as it passes through the sample is illustrated in Figure 1, and plotted as a function of sample thickness.

Figure 1 incorporates data from our recently implemented numerical model [3], illustrating the behavior of electrons at various energies (0.3, 3, 10, 15, 20, 30, and 100 MeV) traversing amorphous ice. Since amorphous ice closely mimics biological environments, this provides insights analogous to those obtained from biological specimens under examination. For sample thicknesses below 10 µm, the projected beam size can be maintained below 10 nm using an electron energy of 10 MeV (black curve in Figure 1a). However, thicker samples up to 20 µm require an electron energy of 30 MeV (red curve in Figure 1a) to achieve a projected beam size below 10 nm. Beyond 30 MeV (green dashed curve in Figure 1a), increasing electron beam energy does not further reduce the maximum size, as geometrical broadening due to electron beam emittance, energy spread, and associated aberrations [3] become more influential than AB effects unless beam emittance can be effectively controlled and further reduced from the optimal value [2,26,27,28].

In contrast, at low energy (300 keV), the beam size broadens much more rapidly, reaching up to 100 nm with a sample thickness of 2.5 µm, as shown in Figure 1b. Our ultimate goal is to establish a mapping of sample thickness to the minimum electron energy required to achieve nanoscale resolution, as shown in Figure 1c.

### 2.2. Monte Carlo Simulation

#### 2.2.1. Using BNL-Monte Carlo Code

To optimize the configuration for a diverse range of sample materials and thicknesses, Monte Carlo (MC) simulations were employed using the framework developed by Wang et al. [29]. The simulation varies the sample thickness from 0 to 20 µm, as well as utilized beam energies of 300 keV, 3 MeV, and 10 MeV. Key parameters, including beam divergence and size at the sample exit, along with the beam size at a detector positioned at 1.5 m from the sample exit ($Z_{det}=1.5\;m$), were systematically studied using a newly implemented MC simulation framework.

Figure 2 illustrates the results, with the top, middle, and bottom rows corresponding to beam energies of 300 keV, 3 MeV, and 10 MeV, respectively. To mitigate the discontinuities and fluctuations in these plots caused by a small number of electrons with large scattering angles, the beam size and divergence are plotted not only in RMS (circles), as is the case for the rest of the manuscript, but also using the corresponding values by including 68% of the incident electrons in the distribution (lines).

For the 300 keV case, the beam divergence and size at the sample exit are depicted in Figure 2a,b, respectively, with the detector beam size depicted in Figure 2c. As sample thickness increases from 1 to 10 µm, the beam size (including 68% electrons) at the detector expands from 29.0 mm to 145.03 mm, necessitating the detector’s closer placement to the sample exit (e.g., 1.5 cm). This increase is mainly due to beam divergence at the sample exit, with the simulated beam size (magenta solid line) aligning closely with the divergence component (magenta dashed line). Divergence rises from 19.33 mrad to 96.68 mrad with sample thickness.

For the 3 MeV case, beam divergence and size at the sample exit are depicted in Figure 2d,e, with the detector beam size shown in Figure 2f. The beam size at the detector increases from 3.8 mm to 19.5 mm as sample thickness increases from 1 to 10 µm, primarily due to divergence at the sample exit. The simulated beam size (blue solid line) closely matches the divergence component (blue dashed line). Divergence increases from 2.5 mrad to 13.0 mrad with sample thickness.

Similarly, for the 10 MeV case, beam divergence and size at the sample exit are illustrated in Figure 2g,h, with the detector beam size shown in Figure 2i. The beam size at the detector increases slightly from 1.5 mm to 1.8 mm as the sample thickness ranges from 1 to 19 µm. As in the 3 MeV case, beam size at the detector is primarily governed by divergence at the sample exit, with the simulated beam size (red solid line) aligning with the divergence contribution (red dashed line). Divergence increases from 1.0 mrad to 1.17 mrad with sample thickness.

Beam divergence in the low-energy 300 keV case increases with sample thickness much more rapidly (see Figure 3a) than in the 3 MeV case (see Figure 3b). Moreover, beam divergence remains nearly constant with increasing sample thickness in the high-energy 10 MeV case, as depicted in Figure 3c. In summary, at the exit of a 10 µm thick sample, there is a strong dependence of beam divergence on beam energy, as illustrated in Figure 3d, which shows beam divergence profiles at three different beam energies: 300 keV (magenta), 3 MeV (blue), and 10 MeV (red). These findings underscore the significant impact of sample thickness on beam characteristics at the detector, emphasizing the crucial role of beam energy-dependent divergence in the design and construction of a MeV-STEM with nanometer resolution.

#### 2.2.2. Comparing BNL-MC Code and GEANT

We conducted a preliminary comparison of simulated beam sizes exiting a 10 μm thick amorphous ice using two MC simulation codes, GEANT [30,31,32] and BNL-MC [29], with an electron beam energy of 10 MeV. Each simulation comprised 10,000 electrons at the focal position, with a root-mean-square (RMS) convergence semi-angle of 1.0 mrad. The beam sizes investigated were 0.5 nm and 1.0 nm for both GEANT and BNL-MC [7,29], totaling four cases.

For an incident beam size of 0.5 nm, both GEANT (Figure 4a) and BNL-MC (Figure 4b) show similar trends in beam size broadening, yielding beam sizes of approximately 10.5 nm at the sample exit. Similarly, for an incident beam size of 1.0 nm, both GEANT (Figure 4c) and BNL-MC (Figure 4d) predict beam sizes of approximately 10.6 nm at the sample exit. Despite the differing initial beam sizes of 0.5 nm and 1.0 nm, both simulation codes consistently predict similar beam sizes.

This consistency can be explained using the analytical model described by Yang et al. [7], where the beam size at the sample exit is expressed as Equation (1)
(1)$$\sigma_{tot}=\sqrt{{\sigma_{AB}}^{2}+{\sigma_{EC}}^{2}}$$

Here, $\sigma_{AB}$ includes both the convergence semi-angle of the incident beam and the scattering broadening during electron-sample interactions [7], and $\sigma_{EC}$ is the beam size at the focal position (top of the sample). $\sigma_{AB}$ is obtained by setting the initial beam size to zero while maintaining a constant convergence semi-angle of 1 mrad, resulting in $\sigma_{AB}=10.5\;nm$ (Figure 4e).

Consequently, for the initial beam sizes of 0.5 nm (Figure 4a,b) and 1.0 nm (Figure 4c,d), the analytical formula (Equation (1)) predicts beam sizes at the sample exit of 10.5 nm and 10.6 nm, respectively. These values closely align with the results obtained from both GEANT and BNL-MC simulations (Figure 4a–d). It is numerically evident that the AB effects play a significantly more important role for a thick biological sample than for a thin (<1 µm) sample in determining the minimum projected beam size.

To illustrate the good agreement between GEANT and BNL-MC, we plot their transverse beam profiles at the sample exit from Figure 4c and d as blue and red curves in Figure 4f, respectively. For an initial beam size of 0.0 nm and a convergence semi-angle of 1 mrad, beam intensity profiles at the exit of the sample with various thicknesses are plotted in Figure 4g. It is important to note that GEANT can employ different physics models; however, in this study, we utilized the default model (standard FTFP_BERT) [33]. Therefore, rigorous validation and comparison of simulation codes necessitate accurate characterizations of beam intensity, size, and divergence.

### 2.3. Numerical Evaluation of the Proposed Methodology

#### 2.3.1. Study Objectives

The aim of this research is to critically evaluate the approximation used by the BNL-MC code for estimating the ratio of the total inelastic scattering cross-section to the total elastic scattering cross-section. This approximation is based on the model proposed by Wolf et al. [8], which is given by:(2)$$R_{in2el}=\frac{\sigma_{inel}}{\sigma_{el}}\approx \frac{\gamma }{Z}\;$$
where *γ* is a parameter approximately equal to 20, and it is relatively independent of the atomic number (Z) or electron energy. This approximation is valid for thin samples where multiple scattering effects are negligible and most high-angle elastic scattering events are accounted for [34]. In simulations, this ratio is typically approximated to ~3 for amorphous ice.

Given that the scattering cross-section quantifies the probability of specific scattering events, this relationship requires validation, especially when multiple scattering occurs. Accurate characterization of angular broadening, as illustrated in Figure 3, as a function of sample thickness and composition, is crucial. Moreover, energy-resolved angular broadening characterizations can provide important additions to differentiate elastic and inelastic scattering processes, which occur without and with energy losses, respectively. This can be achieved by repeating the following procedure with an activated zero-energy filter (a combination of a dipole magnet and downstream aperture) [11]. This energy filter only allows the portion of un-scattered and elastically-scattered electrons to reach the detector. According to our early study [7], the ratio of critical angles of elastic and inelastic scattering stays nearly constant within the electron energy range of 1–10 MeV, around 12.9. In the zero-energy filter scenario, elastic scattering events exhibit significantly larger scattering angles. Consequently, these measurements will enable a precise extrapolation of these critical angles.

We anticipate that these investigations will yield detailed information on critical electron scattering angles as functions of electron beam energy and sample composition. These data are crucial for refining the parameters used in MC simulations and improving their predictive accuracy. Consequently, we aim to establish a mapping between sample thickness and the required minimum electron energy (see Figure 1c), which could lead to more cost-effective designs of MeV-STEM instruments.

#### 2.3.2. Key Parameters for Measurement

The study aims to measure three key parameters:Beam divergence: We will assess the divergence angle of the electron beam by varying the detector’s longitudinal position relative to the sample exit, analyzing how the beam profile changes with distance. Additionally, by steering the beam across a wedge-shaped sample with thicknesses ranging from 0 to 20 μm (see details in Section 2.3.5), we will quantify beam spread as a function of sample thickness.Beam intensity: We will measure electron beam attenuation through various sample thicknesses and materials by correlating these measurements with incident beam intensity.Beam size at sample exit: Direct measurements of the projected beam size on the detector will be obtained from the 2-D image of the electron distribution. Using the measured beam divergence (details in the next section), we will calculate the beam size at the sample exit, as described by Equation (3) in Section 2.3.4.

#### 2.3.3. Detector Arrangement

Positions: The detector will be positioned at 0.2 m, 0.5 m, and 1.5 m from the sample exit for several MeV or higher energy levels (Figure 5a).Optimization: These positions were chosen based on simulations and previous studies to achieve optimal linear fits for accurate measurements of beam divergence and size.Constraints:○Minimum distance (Z_det,min_): Ensures that the beam size on the detector is sufficiently large for reliable measurements, considering pixel size and detector array dimensions.○Maximum distance (Z_det,max_): Prevents the beam from becoming too large to measure accurately.

The setup is designed to accurately measure beam divergence and size at the sample exit. Simulations using amorphous ice at detector positions of 0.2, 0.5, and 1.5 m, with a collection angle of *β* = 50 mrad, cover electron energies from 1 to 10 MeV with a 10 μm thick sample, facilitating the extraction of beam divergence and size. For low-energy scenarios, such as those involving 300 keV, measurements may need to be conducted at a different facility, assuming that 300 keV STEM microscopes are widely accessible. Reliable beam size measurements necessitate a beam size range of 0.1 to 30 mm (as indicated by the green-highlighted area in Figure 5b). Consequently, the detector should be positioned no more than 20 cm from the sample exit to ensure measurement accuracy. Additionally, adjustments may be necessary when working with materials like silicon, as varying densities can affect beam divergence and the slope of the measurements.

#### 2.3.4. Detector Performance Simulation

Simulations were carried out to validate the optimal positions of the detectors. The key findings are as follows:Placing the detector closer than 0.2 m from the sample exit does not improve the precision of divergence measurements in high-energy cases (1–10 MeV).The selected positions at 0.2 m, 0.5 m, and 1.5 m offer optimal conditions for accurate measurements of beam divergence and size. However, if feasible, positioning the detector closer to the sample exit could further enhance the precision of beam size measurements.

Simulations used amorphous ice with a thickness of 10 μm and electron energies of 3 MeV (Figure 6, right) and 10 MeV (Figure 6, left). A three-detector configuration was employed at distances of 0.2 m, 0.5 m, and 1.5 m, and additional positions were tested at 0.05 m and 0.1 m. The achieved precision was 1% for divergence and 13% for beam size at 3 MeV; and 0.02% for divergence and 12% for beam size at 10 MeV (Table 1). The term ‘Fitting detector data’ refers to propagating the particle distribution from the sample exit to the detectors located at 0.2 m, 0.5 m, and 1.5 m. Measurements are fitted linearly to derive divergence and beam size from the slope and residuals. Further positioning of detectors within 0.2 m did not improve measurement precision. According to Equation (3),
(3)$$\sigma_{det}\left(z\right)=\tan{(\sigma }_{\theta ,\;z=0})\cdot z+\sigma_{r,z=0}$$
precision benefits from increased distance due to the angular divergence ${(\sigma }_{\theta ,\;z=0}$). Therefore, expanding to a five-detector setup did not enhance accuracy. While closer detector placement could enhance beam size precision ($\sigma_{r,z=0}$), distances under 0.01 m may be impractical due to the very small beam sizes (e.g., <0.1 µm at 10 MeV).

The estimates of measurement precision for beam size and divergence represent worst-case scenarios. If we assume that the electron beam is focused to a spot size greater than 1 μm at the sample entrance, similar to the conditions observed at the PEGASUS beamline [11,12,13], we can improve the precision of beam size measurements to within a few percent. Since beam divergence is primarily influenced by the AB effect, we anticipate that the precision of measuring beam divergence at the sample exit will match or exceed that of the variable detector configuration, achieving an accuracy of 1% or better.

#### 2.3.5. Numerical Assessment of Sample Fabrication

In this study, we use a silicon wafer as an example to demonstrate that the methodology for fabricating a wedge-shaped sample is applicable to various materials. The analysis will utilize a wedged silicon sample with a thickness ranging from 0.0 to 20.0 μm. The metrology group at NSLS-II has a silicon wafer with dimensions of 30 mm × 10 mm × 20 μm. To fabricate the wedged sample, we will employ the focused ion beam with shadow technique. Silicon is commonly used as a substrate material for microchips, while carbon, a low-Z material, closely resembles biological materials.

The simulation has been conducted with the following parameters:The process begins with the silicon substrate.Achieving the desired slope of 0.67 μm/mm in the x-direction requires approximately 3.3 h, as illustrated in Figure 7.

This wedged silicon sample is a promising candidate for this standardized methodology. To ensure sample integrity and consistency, the same sample holder will be used for both fabrication and measurement, thereby eliminating the need for transportation, which is crucial given the fragile nature of the sample.

### 2.4. Summary of Key Issues

The objective of these simulations is to develop a standardized methodology to investigate how electron–sample interactions affect imaging resolution across various materials and sample thicknesses. This study can provide detailed information on critical scattering angles of electrons as functions of beam energy and sample composition, which is essential for refining MC simulation parameters and enhancing their predictive accuracy.

For possible setups, a high-energy electron accelerator, such as PEGASUS [11] or the system proposed by Yang et al. [2], will be utilized. Wedged-shaped samples will cover a wide range of thicknesses, and adaptable detector configurations will be implemented to accommodate various materials and thicknesses.

The expected outcomes include the validation of analytical models and MC simulations to improve predictive accuracy, as well as the optimization of electron microscopy techniques to enhance imaging resolution and accuracy, particularly focusing on thick biological samples and microchip examinations.

## 3. Conclusions

Numerical simulations of scattering processes in electron microscopy often encounter computational limitations that can prevent accurate modeling of electron interactions with biological and microchip samples. These simulations frequently fall short of capturing the full complexity of real systems. Therefore, validation through empirical observations is essential for refining theoretical models and revealing unforeseen phenomena, ultimately enhancing our understanding of electron scattering. Rigorous assessments are crucial for benchmarking and improving numerical tools, enabling advancements beyond the current limitations of simulations.

Our simulation study suggests that this methodology could effectively validate the angular broadening effects predicted by both analytical and numerical models for electrons interacting with thick biological samples and microchip materials. The flexible design of the methodology will facilitate precise characterization across a wide range of materials and sample thicknesses within a unified setup. These efforts are vital for the development of advanced MeV-STEM/TEM instruments, which can achieve nanoscale resolution for bio-sample imaging without the need for sample slicing while also providing detailed information for microchip defects analysis. Success in this endeavor could significantly accelerate UEM imaging using MeV-STEM/TEM, potentially reducing imaging times by more than tenfold compared to current methods [3,4,5,34]. Additionally, the initiative aims to address uncertainties associated with the FIB process [6,35,36,37,38], presenting a more efficient and precise imaging technique for biological samples and microchip engineering.

Key questions to be explored include:How do electron interactions with biological samples and microchip materials vary with beam energy, particularly concerning angular broadening? The standardized methodology will facilitate the precise determination of these critical angles.How does altering the electron bunch structure through the drive laser system affect radiation damage in biological samples, considering variation in bunch structure, energy, and intensity (see Figure A4 in the Section A.4)?

A comparative analysis of MC simulations, such as GEANT and BNL-MC codes, indicates consistency in angular broadening effects; however, discrepancies in magnitude may arise with different electron beam energies. Notably, higher beam energies could lead to increased operational costs. Therefore, measurements of beam size and divergence are critical for rigorously evaluating and optimizing imaging parameters.

These questions highlight the indispensable role of empirical observations in validating theoretical models and simulation outputs, which is essential for advancing MeV-STEM/TEM instrumentation tailored for the precise imaging of biological samples and detailed microchip defect analysis.

## Figures and Tables

**Figure 1 nanomaterials-14-01797-f001:**
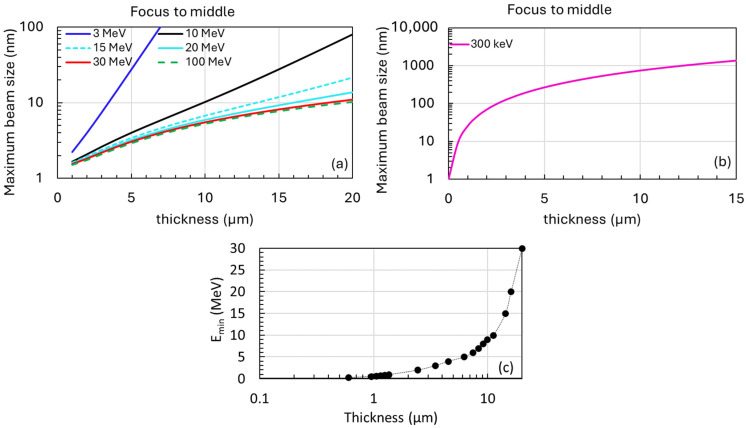
The maximum beam size of the probe when the electron beam is focused at the midpoint of the amorphous ice sample thickness as a function of the sample thickness. (**a**) The plot includes data for six different electron energies: 3 MeV (blue), 10 MeV (black), 15 MeV (cyan dash), 20 MeV (cyan), 30 MeV (red), and 100 MeV (green dash). (**b**) An identical plot for lower electron energies of 300 keV (magenta). (**c**) To maintain a projected beam size below 10 nm (the optimal resolution), we plot the required minimum electron energy as a function of sample thickness. This analysis assumes a constant emittance of 2 picometers across the electron energy range of 1 to 10 MeV.

**Figure 2 nanomaterials-14-01797-f002:**
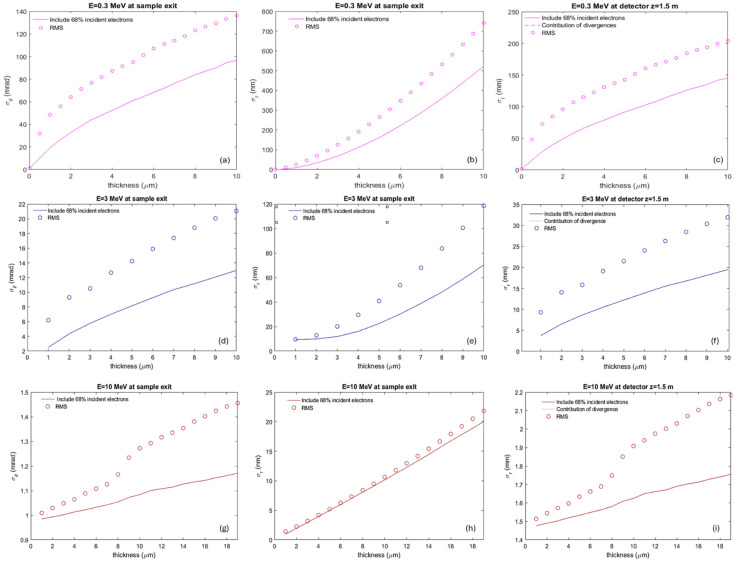
MC simulation results depicting beam divergence and size at the sample exit, as well as beam size at the detector ($Z_{det}=1.5\;m$), as functions of sample thickness for beam energies of (**a**–**c**) 300 keV, (**d**–**f**) 3 MeV, and (**g**–**i**) 10 MeV. Panels (**a**,**d**,**g**) show beam divergence; panels (**b**,**e**,**h**) display beam size at sample exit; and panels (**c**,**f**,**i**) depict beam size at the detector. Solid lines represent simulated beam sizes, while dashed lines indicate contributions from beam divergence at the sample exit. Beam size and divergence are plotted with RMS values (circles) and for 68% of the incident electrons (lines).

**Figure 3 nanomaterials-14-01797-f003:**
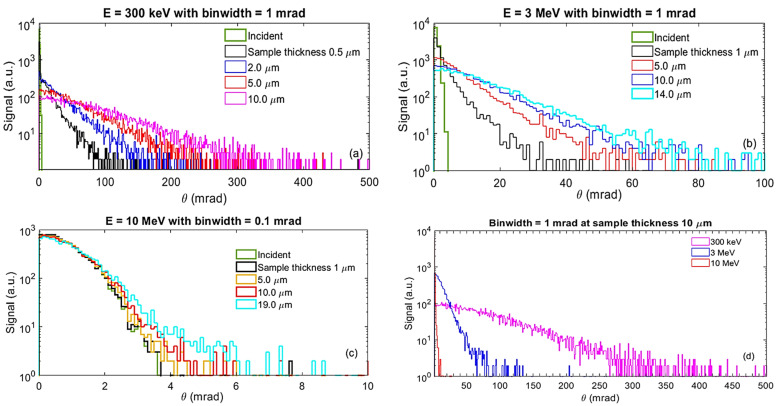
Panels (**a**), (**b**), (**c**) present beam divergence profiles at various sample thicknesses for beam energies of 300 keV, 3 MeV, and 10 MeV, respectively. The bin width was reduced from 1 mrad for the 300 keV and 3 MeV cases, and to 0.1 mrad for the 10 MeV case. Additionally, the range of beam divergence on the *x*-axis was reduced from 0 to 500 mrad for the 300 keV case, to 0–100 mrad for the 3 MeV case, and 0–10 mrad for the 10 MeV case. Panel (**d**) displays beam divergence profiles at the sample exit with a thickness of 10 µm for beam energies of 300 keV (magenta), 3 MeV (blue), and 10 MeV (red), respectively.

**Figure 4 nanomaterials-14-01797-f004:**
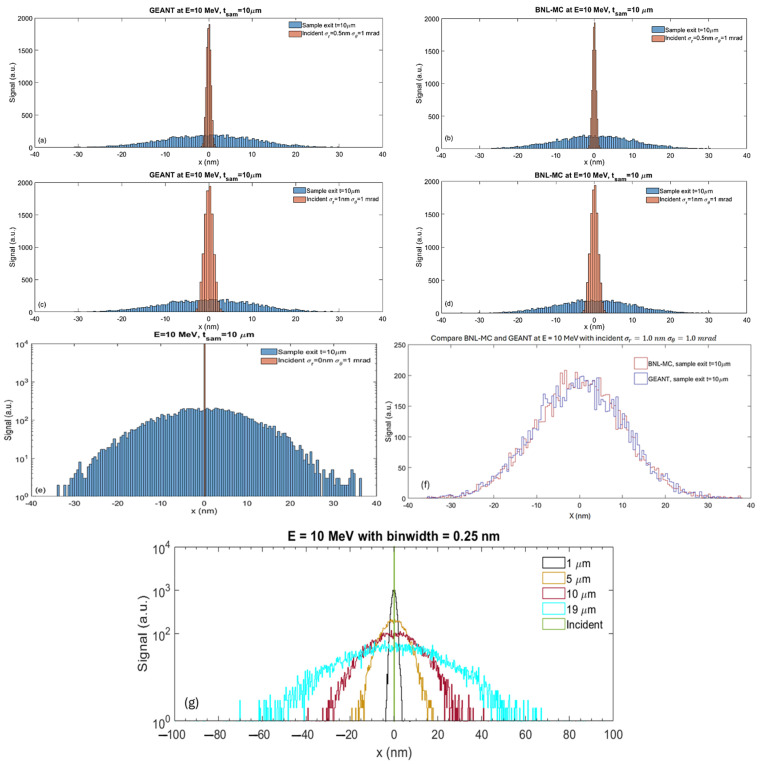
Electron beam energy at 10 MeV. In each plot of (**a**–**e**), orange and blue curves represent beam intensity profiles at the entrance and exit of water (analogous to amorphous ice) with a thickness of 10 µm, respectively. The specifics of each simulation are as follows: (**a**) GEANT simulation, with an initial beam size of 0.5 nm and a convergence semi-angle of 1.0 mrad. (**b**) BNL-MC simulation, with an initial beam size of 0.5 nm and a convergence semi-angle of 1.0 mrad. (**c**) GEANT simulation, similar to (**a**), but with an incident beam size of 1.0 nm. (**d**) BNL-MC simulation, similar to (**b**), but with an incident beam size of 1.0 nm. (**e**) MC simulation (similar for GEANT and BNL-MC), with an initial beam size of 0.0 nm and a convergence semi-angle of 1.0 mrad; log in y is used for plotting sample incident and exit beam profiles on the same scale. In all plots except (**a**,**b**,**e**), the bin widths are 0.5 nm. For (**a**,**b**,**e**), the incident beam profile is plotted with a bin width of 0.25 nm. (**f**) Transverse beam profiles at the sample exit from (**c**,**d**) are plotted as blue and red curves, respectively. (**g**) With an initial beam size of 0.0 nm and a convergence semi-angle of 1 mrad, transverse beam profiles are plotted at various sample thicknesses, 1 μm (black), 5 μm (orange), 10 μm (purple), and 19 μm (cyan), respectively.

**Figure 5 nanomaterials-14-01797-f005:**
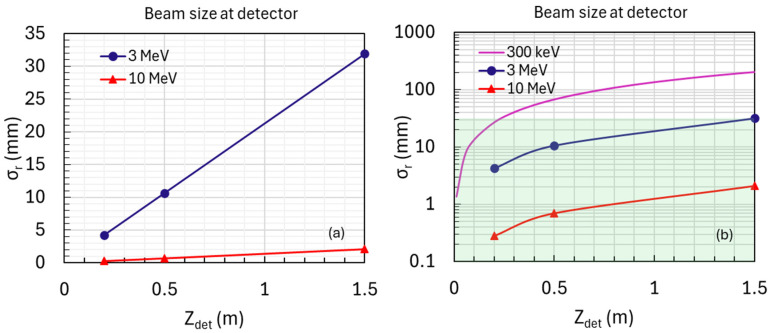
(**a**) Detector signals measured for the electron beam passing through amorphous ice at detector positions *Z*_det_ = 0.2, 0.5, and 1.5 m. Beam size measurements are taken at a collection angle of *β* = 50 mrad, with electron energies of 3 MeV (blue) and 10 MeV (red), using a fixed sample thickness of 10 μm. (**b**) Similar to (**a**), with the addition of a low-energy 300 keV case. The highlighted green region indicates the range where reliable beam size measurements are obtained.

**Figure 6 nanomaterials-14-01797-f006:**
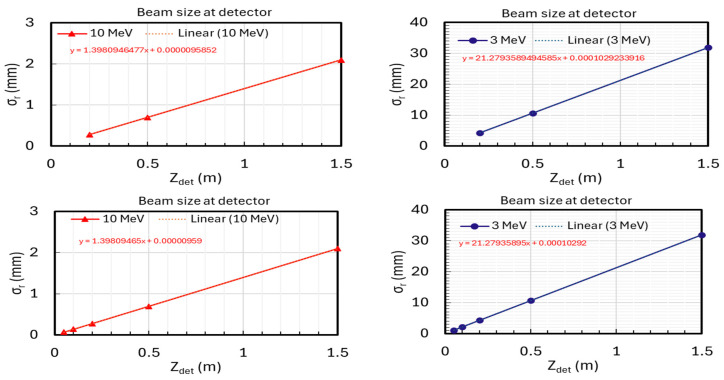
The top and bottom rows show configurations with a 3-detector ($Z_{det}$ = 0.2, 0.5, and 1.5 m) and a 5-detector ($Z_{det}$ = 0.05, 0,1, 0.2, 0.5, and 1.5 m) setup. The left and right columns represent electron beam energies of 10 MeV and 3 MeV, respectively. Linear fittings from both configurations yield identical slopes and residuals, indicating no improvement in precision with the additional detectors.

**Figure 7 nanomaterials-14-01797-f007:**
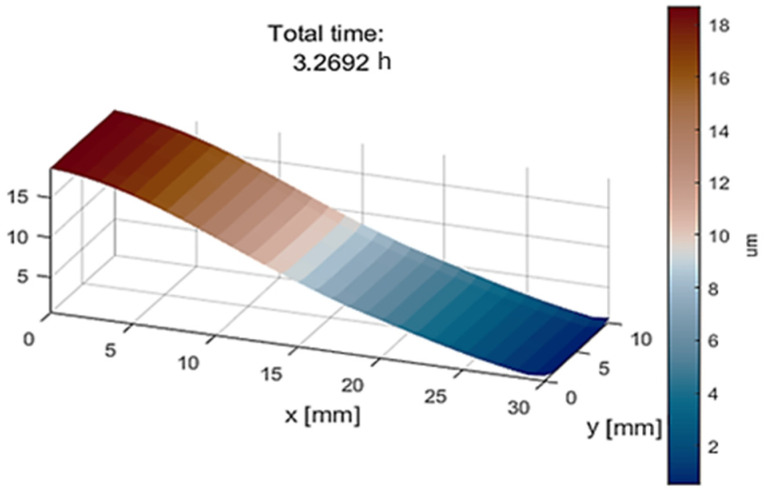
Simulation confirms that it takes 3.3 h to achieve the slope of (20 µm)/(30 mm) = 0.67 µm/mm using the focusing ion beam with shadow technique.

**Table 1 nanomaterials-14-01797-t001:** Linear fitting of detector signals yields slopes corresponding to the divergence angle and residuals corresponding to beam size. Differences between fitted and simulated values estimate systematic measurement errors.

	E	Divergent Angle	Beam Size	E	Divergent Angle	Beam Size
		mrad	nm		mrad	nm
MC-simulation	10 MeV	1.397794	10.847086	3 MeV	21.073517	118.607333
Fitting detector data		1.398095	9.585200		21.279359	102.923392
Measurement error (%)		0.021504	11.633414		0.976779	13.223416

## Data Availability

The datasets generated and analyzed during the current study are not publicly available due to the reason that we want to know who has an interest in our datasets but are available from the corresponding author upon reasonable request.

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
