# Peer review of "Simulation Study of High-Precision Characterization of MeV Electron Interactions for Advanced Nano-Imaging of Thick Biological Samples and Microchips"

_nanomaterials, 2024, doi:10.3390/nano14221797_

Round 1

Reviewer 1 Report (Previous Reviewer 1)

Comments and Suggestions for Authors

The way the paper is written in the revised version is still misleading in my opinion. Already in the abstract “Precise measurements of these parameter” is mentioned as if experiments have been done. Moreover, in the first paragraph of the conclusions the value of measurements is emphasized followed by “This planned experiment aims to validate” still looking more like a proposal then a research paper. I suggest to rewrite the paper considering the simulation as the experiment.

The authors have now included 300keV simulations but I belief it should be possible to conduct this experimentally because there are plenty 300keV STEM microscopes in the field, this will give the validation of the model the authors are looking for.

I suggest to leave out the experimental procedure completely and focusing on the simulations.

Please add “simulation” to the title.

Comments on the Quality of English Language

many sentences need a rereading

Author Response

Dear referee, 

We appreciated you for spending valuable time reviewing our paper and providing highly constructive advice and suggestions. We have implemented all the recommended changes based on the referee’s suggestions, which have greatly improved the quality of our manuscript. Most importantly, to avoid the misleading ‘as if the experiments have been done’, we clearly indicate all the studies were conducted in simulations.

Thank you very much.

Reviewer 2 Report (New Reviewer)

Comments and Suggestions for Authors

I do not have comments to do, I think the paper could be accepted in the present form.

Author Response

Dear referee, 

We appreciated the reviewer spending valuable time reviewing our paper and providing highly constructive advice and suggestions.

We have made additional changes, which could address the referee’s concerns better than before. 

Thank you very much.

Round 2

Reviewer 1 Report (Previous Reviewer 1)

Comments and Suggestions for Authors

In the revised version of “Simulation Study of High-Precision Characterization of MeV Electron Interactions for Advanced Nano-Imaging of Thick Biological Samples and Microchips” a little progress has been made to focus more on the simulations rather than the proposed experiments. However, the introduction is still full of annotations to a proposal rather than a scientific paper and needs therefore restructuringsentences like below need attention.

 “In our proposed experimental procedure utilizing an electron accelerator, such as PEGASUS at UCLA [14-16], we aim to investigate a wide range of beam energies from….” Why not, Here we simulated a wide range of beam energies….? In this way there cannot be any confusion about the this study

Our study will involve silicon and carbon samples…..better is our study involves …..

“Our primary focus will be on detecting the transmitted electron intensity profile and analyzing angular distribution”.  Better would be,  here we simulated the transmitted electron intensity profile……

“In designing an experiment through simulation studies to determine the optimal electron beam energy for an MeV-STEM instrument…” better would be, in these simulations the optimal electron beam energy…..

Section 2.3.2 only describes a planned experiment. Please remove this and publish when the experiment is done

Author Response

Dear referee:

We greatly appreciated you for dedicating valuable time to reviewing our paper and providing highly constructive feedback and suggestions.

We have done our utmost to address your requirements while keeping the core research motivation intact. Most importantly, to avoid any misunderstandings, we clearly indicate all studies were conducted through simulations and have eliminated all potentially confusing terms related to ‘experiment’, ‘experimental’, and ‘proposal’ (see the marked manuscript and line-by-line response).

Thanks again for your valuable time.

Round 3

Reviewer 1 Report (Previous Reviewer 1)

Comments and Suggestions for Authors

This paper looks much better this way, at least like a research paper. However I doubt if ever a 10meV microscope will be build. Making lamellae is now a standard practice for high resolution TEM in both biology and material science.

Author Response

Dear reviewer,

We greatly appreciated the reviewer for spending valuable time reviewing our paper and providing highly constructive advice and suggestions. The line-by-line responses are as follows:

However I doubt if ever a 10meV microscope will be build.

We understood the reviewer’s skepticism. However,  as outlined with details in the Introduction,  

“…Recent advancements in photocathode guns and superconducting radio frequency (SRF) cavities suggest that constructing an MeV ultrafast electron microscope (UEM) could be feasible within a reasonable budget. Prior simulation studies and ongoing MeV-UEM hardware development indicate that essential components, such as ultra-low emittance photocathode guns, SRF accelerating cavities, and momentum apertures, have already been successfully demonstrated [2]. ..….”, further details are available in reference [2], Towards Construction of a Novel Nanometer-resolution MeV-STEM for Imaging of Thick Frozen Biological Samples. Photonics 11(3), 252, doi:10.3390/photonics11030252 (2024).  

Additionally, the UK’s Relativistic Ultrafast Electron Diffraction and Imaging (RUEDI) center, led by the University of Liverpool, is currently constructing a 4 MeV electron microscope to drive advancements in ultrafast electron diffraction and imaging.

Making lamellae is now a standard practice for high resolution TEM in both biology and material science.

We have thoroughly addressed various volume electron microscopy techniques and their associated challenges in reference [2]. Additionally, we added a few sentences describing the artifacts related to lamella preparation in the first paragraph of Introduction, as follows:

“…... …. [3-6]. Moreover, charging artifacts from lipid deposits, curtaining effects from density variations, and linear artifacts from the milling process add significant complexity to imaging. Thus, rapid, efficient imaging of thick samples at nanoscale resolution is essential for advancing scientific discovery [2]….”

Thank you very much, -xi

This manuscript is a resubmission of an earlier submission. The following is a list of the peer review reports and author responses from that submission.

Round 1

Reviewer 1 Report

Comments and Suggestions for Authors

The paper “Development of Experimental Procedure to Characterize MeV-Electron Sample Interactions toward Nano-Imaging of thick Biological Samples and Microchips” describes the possible experiments on the interaction of mega volt electrons with matter, but unfortunately there were no experiments done yet. The paper looks more like a research proposal for funding rather than a scientific paper. The topic is interesting because there is still little known on the effect of electron scattering (inelastic and multiple scattering) in relation to the thickness of the sample and the electron energy. However, modern electron microscopy has abandoned the idea that higher energy is better and is focusing on 200-300 keV electron energy. For structural biology the trend is even to go to 100 keV because of the costs of running a 300 keV machine. Expanding the focus of this research into lower energy would be more than welcome also because there is no longer a need to image thicker objects with medium resolution ~10nm-50nm because this is now typically done with super resolution light microscopic techniques (Nobel prize 2014) and widely available nowadays.  For single particle cryo electron microscopy the thickness is one of the most crucial parameter upon recording large data sets because already at ice thicknesses above 50 nm the resolution tends to go up due to beam broadening and multiple scattering events in a range 100 to 300keV and maybe 1meV.

Reviewer 2 Report

Comments and Suggestions for Authors

In their paper, Yang et al. propose an experimental setup to measure the intricate electron scattering and beam broadening effects of high-energy electrons in the 3-100 MeV range as they travel through several µm thick specimens, such as biological samples (ice) or microelectronic devices (silicon). They performed a series of Monte Carlo (MC) simulations to assist in the experimental design and explore different settings across a wide parameter space. While the presented results are interesting and certainly deserve publication in this special issue, there are a few general questions that I suggest the authors discuss in more depth to make their proposal more accessible to a broader readership:

- The authors aim to contribute to the advancement of imaging resolution and accuracy in electron microscopy of biological samples and defects in microchips. What does this entail in detail? Where are the current limits, and what could be achieved with a better understanding of the underlying scattering processes? What specific research questions could be addressed through such advancements? What types of biological samples or defects could be analyzed that are currently inaccessible? Can the authors provide specific examples?

- What are the limitations of the simulations? Where do the authors expect to gain new insights from their experiments? Why is it crucial to perform additional experiments?

- I find the idea of manipulating the electron bunch structure to minimize beam damage effects particularly interesting. It is worth mentioning that beam damage effects are a fundamental limitation, especially with MeV electrons, which makes it challenging to obtain images such as those shown in Fig A1c.